

# Dendrobium alkaloids decrease Aβ by regulating α- and β-secretases in hippocampal neurons of SD rats

Juan Huang[1,*], Nanqu Huang[2,*], Minghui Zhang[3], Jing Nie[1], Yunyan Xu[1], Qin Wu[1] and Jingshan Shi[1]

[1] Key Laboratory of Basic Pharmacology and Joint International Research Laboratory of Ethnomedicine of Ministry of Education, Zunyi Medical University, Zunyi, China
[2] Drug Clinical Trial Institution, The Third Affiliated Hospital of Zunyi Medical University, The First People's Hospital of Zunyi, Zunyi, China
[3] Tongren People's Hospital, Tongren, China
[*] These authors contributed equally to this work.

## ABSTRACT

**Background**. Alzheimer's disease (AD) is the primary cause of dementia in the elderly. The imbalance between production and clearance of amyloid β (Aβ) is a very early, often initiating factor in AD. *Dendrobium nobile* Lindl. alkaloids (DNLA) extracted from a Chinese medicinal herb, which have been shown to have anti-aging effects, protected against neuronal impairment *in vivo* and *in vitro*. Moreover, we confirmed that DNLA can improve learning and memory function in elderly normal mice, indicating that DNLA has potential health benefits. However, the underlying mechanism is unclear. Therefore, we further explored the effect of DNLA on neurons, which is closely related to learning and memory, based on Aβ.

**Methods**. We exposed cultured hippocampal neurons to DNLA to investigate the effect of DNLA on Aβ *in vitro*. Cell viability was evaluated by MTT assays. Proteins were analyzed by Western blot analysis.

**Results**. The cell viability of hippocampal neurons was not changed significantly after treatment with DNLA. But DNLA reduced the protein expression of amyloid precursor protein (APP), disintegrin and metalloprotease 10 (ADAM10), β-site APP cleaving enzyme 1 (BACE1) and $A\beta_{1-42}$ of hippocampal neurons in rats and increased the protein expression of ADAM17.

**Conclusions**. DNLA decreases Aβ by regulating α- and β-secretase in hippocampal neurons of SD rats.

Corresponding author
Jingshan Shi, shijs@zmu.edu.cn

## INTRODUCTION

Alzheimer's disease (AD) is a neurodegenerative disease of the central nervous system that causes dementia in a large percentage of the aged population and for which there are only symptomatic treatments. Clinically, AD is typically characterized by progressive loss of memory, declining cognitive function, decreasing physical function and ultimately death. The biology of AD is characterized by two major protein abnormalities in the
brain of affected individuals: the extracellular accumulation of amyloid β (Aβ) plaques and intraneuronal deposits of neurofibrillary tangles (NFTs) (*De-Paula et al., 2012*). At present, the pathogenesis of AD has yet to be fully elucidated, and the main hypotheses proposed include deposition of Aβ protein, loss of choline neurons, abnormal activation of inflammatory reactions, disturbance of energy metabolism, genetic abnormalities and oxidative stress (*Choudhry et al., 2012*; *Mouton-Liger et al., 2012*). In addition, environmental factors, diet and diseases increase the risk of AD (*Ashok et al., 2015*; *Banerjee et al., 2014*; *Cansu et al., 2017*; *Cheng et al., 2015*; *Faraco et al., 2016*; *Yousof Ali, Jung & Choi, 2015*). The Aβ peptide cascade plays a critical role in the development of AD (*Baranello et al., 2015*).

Aβ is generated by amyloid precursor protein (APP) metabolism. APP in its mature form can be processed by at least two proteolytic pathways (*Kowalska, 2004*), the so-called "non-amyloidogenic" and the "amyloidogenic" pathways. In the first pathway, α-secretases cleaves APP with the γ-secretases, thus impeding the formation of the toxic Aβ peptide. In the second pathway, Aβ, which mainly consists of the $Aβ_{1-40}$ and $Aβ_{1-42}$ peptides, is produced in a two-step proteolytic process initiated by β-secretase and next mediated by γ-secretase. In brief, α-, β- and γ-secretases play key roles in the metabolic processing of APP. Strategies to reduce the formation of toxic Aβ are an obvious approach to prevent AD. Activation of α-secretase promotes the cleavage of APP, which can reduce the production of Aβ (*Lichtenthaler, 2012*). And the inhibition of β-secretase (*Hilpert et al., 2013*) and γ-secretase (*Wolfe, 2012*) activity can also reduce the production of Aβ. Therefore, targeting α-, β- and γ-secretases is a promising focus of AD research.

*Dendrobium* (*D.*) *nobile* Lindl. is a traditional Chinese herbal medicine and medicinal material of Guizhou Province. The chemical constituents of *D. nobile* Lindl. include alkaloids, sesquiterpene, bibenzyl, fluorenone, phenolic acid, phenylpropanoid, phenanthrene, polysaccharides, and lignans. An ancient medical book notes that this treatment can strengthen the body and extend life. Most compounds had good physiological activity, could significantly improve memory loss and cerebral ischemia, and showed anti-fatigue, antioxidative, blood sugar lowering, antitumor and anti-inflammatory effects. As a pharmacologically active ingredient of *D. nobile* Lindl., *D. nobile* Lindl. alkaloids (DNLA) was originally extracted from *D. nobile* Lindl and has significant protective effects in the nervous system. The chemical structures of DNLA are shown in the Fig. 1, and the chromatograms of the sample solutions are shown in Fig. 1B (*Nie et al., 2016*). Previous studies indicated that DNLA can improve the neuronal disruption caused by lipopolysaccharide (*Li et al., 2011b*), oxygen-glucose deprivation (*Wang et al., 2010*) and reperfusion and decrease neuronal apoptosis, hyperphosphorylation of tau protein, and Aβ deposition in the rat brain (*Nie et al., 2016*). Furthermore, we observed amelioration of the spatial learning performance in AD model rats induced by $Aβ_{25-35}$, and this effect may be related to decreases in the generation of $Aβ_{1-42}$ (*Zhang et al., 2016*) and alleviation of $Aβ_{25-35}$-induced axonal injury by improving autophagic flux in neurons *in vitro* (*Li et al., 2017*; *Zhang et al., 2017*). Notably, DNLA improved learning and memory function in elderly normal mice (*Linshan et al., 2016*). This caught our attention and we really want to know why there is such an improvement.

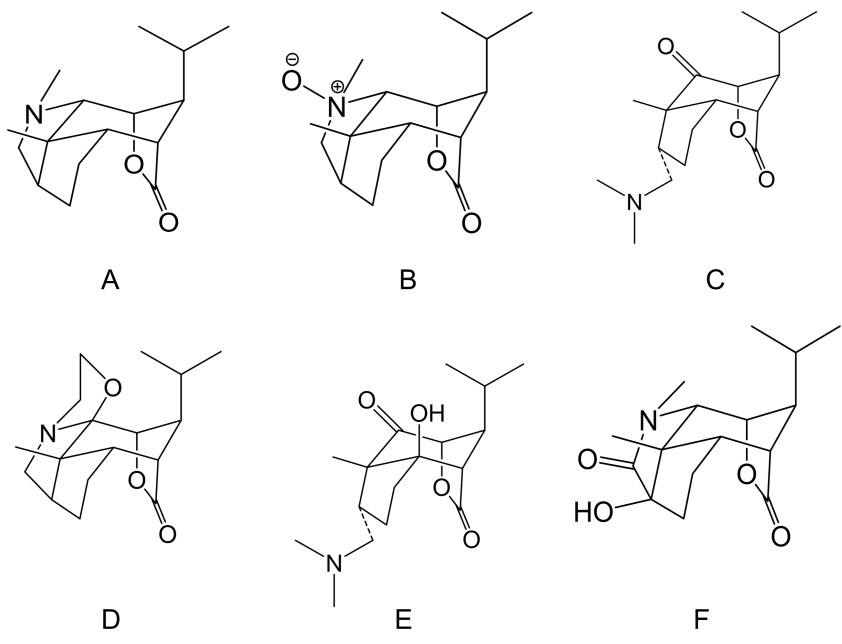

**Figure 1** **Chemical structures of *Dendrobium nobile* Lindl alkaloids.** (A) Dendrobine, (B) Dendrobine-N-oxide, (C) Nobilonine, (D) Dendroxine, (E) 6-Hydroxy-nobilonine, and (F) 13-Hydroxy-14-oxodendrobine.

Therefore, based on the background of AD and our previous works, we designed the following experiment that aims to investigate the effect of DNLA on Aβ and related secretases in the normal hippocampal neurons of rats and to further explore the mechanism underlying the regulation of the APP metabolic pathway.

## MATERIALS & METHODS

### Materials

Dendrobium was purchased from Xintian Traditional Chinese Medicine Industry Development Co., Ltd., of Guizhou Province. DNLA was isolated from the extracts and analyzed by LC MS/MS. Alkaloids accounted for 79.8% of the DNLA and were comprised of 92.6% dendrobine ($C_{16}H_{25}O_2N$), 3.3% dendrobine-N-oxide ($C_{16}H_{25}O_3N$), 2.0% nobilonine ($C_{17}H_{27}O_3N$), 0.9% dendroxine ($C_{17}H_{25}O_3N$), 0.32% 6-hydroxy-nobilonine ($C_{17}H_{27}O_4N$), and 0.07% 13-hydroxy-14-oxodendrobine ($C_{16}H_{23}O_4N$) (*Nie et al., 2016*). Trypsin (SH30042.01) and DMEM/F12 medium (SH3002301B) were purchased from HyClone (America). FBS (S9030) was purchased from Solarbio (China). DES, Neurobasal A (A2477501), and B27 (17504044) were purchased from Gibco (America). The goat anti-mouse IgG H&L (Alexa Fluor® 488) was purchased from Life Technologies (America). Anti-beta III tubulin antibody (ab14545), goat anti-rabbit IgG H&L (ab150113), anti-disintegrin and metalloprotease (ADAM)10 (ab1997), anti-β-site APP cleaving enzyme 1(BACE1) (ab2007), anti-presenilin 1(PS1) (ab76083), and anti-Aβ$_{1-42}$ (ab201060) were purchased from Abcam (America). Anti-APP (D260097) and anti-ADAM17 (D151531) were purchased from Sangon Biotech (China).

## Animals

Sprague-Dawley (SD) rats (approximately 200~250 g) were purchased from the Laboratory Animal Center, Chongqing, China [grade: specific pathogen-free (SPF), certificate no. SCXK 2012-0005] and housed at 22~23 °C with a 12-hour light/dark cycle. Five SD rats (male: female = 1:4) were housed in each cage to propagate the newborn SD rats and had free access to food and water. All procedures were undertaken with the approval of the Animal Experimental Ethical Committee of Zunyi Medical University [No. (2019)2-995, 11 Mar 2019].

## Pretreatment of DNLA

DNLA was soluble in dimethyl sulfoxide (DMSO) and was stored at $-20$ °C. When used, DNLA was diluted to different concentrations with Neurobasal A supplemented with 2% B27. The final concentration of DMSO was 0.01% (v/v) in our experimental system.

## Culture of hippocampal primary neurons and identification

Hippocampal tissues from newborn SD rats born within 24~48 h were separated on ice and then incubated in 0.125% trypsin at 37 °C with 5% $CO_2$ for 15 min. Hippocampal tissues were triturated by passing repeatedly through a 1-mL pipette tip and filtered through a sieve with 200, 300, and 400 mesh. Then, the cells were collected by centrifugation at 179 g for 8 min. Cells were seeded on poly-L-lysine-coated (four mg/mL) 6-well, 24-well plates or 96-well plates and cultured in DMEM/F12 medium supplemented with 10% FBS, 10% DES, 100 U/mL penicillin and streptomycin at 37 °C with 5% $CO_2$. After 4 h, DMEM/F12 medium was replaced with Neurobasal A supplemented with 2% B27 for the duration of the experiments. Half of the liquid was exchanged every two or three days. On the 3rd day, neurons were treated with cytarabine. On the 8th day, neurons in good condition were used for the following experiments. The animal procedures were approved by the Animal Experimentation Ethics Committee of Zunyi Medical University. The profile of neurons was visualized by immunofluorescence staining using mouse monoclonal anti-beta III tubulin antibody (1:1,000) and goat anti-mouse IgG H&L (1:1,000). DAPI (1:20) was used to mark the cell nucleus.

## Assessment of cell viability by MTT assays

Neurons were seeded into 96-well plates and treated with DNLA for a desired time period at the indicated concentrations. Five replicates were made for each treatment. After treatment, cell viability was evaluated by MTT assays as previously described. The cell viability is expressed as a percentage of the OD of cells with the indicated treatments to that in cells treated with the DMSO control.

## Western blot assay

Total protein was extracted from cultured neurons using a total protein extraction kit and quantified by a BCA protein assay kit. Equal amounts of protein (20 μg) per lane were separated by SDS-PAGE gels and then transferred to a PVDF (0.45 μm) membrane. The membranes were incubated with the following primary antibodies: anti-APP (1:500), anti-ADAM10 (1:1,000), anti-ADAM17 (1:1,000), anti-BACE1 (1:1,000), anti-PS1 (1:1,000),

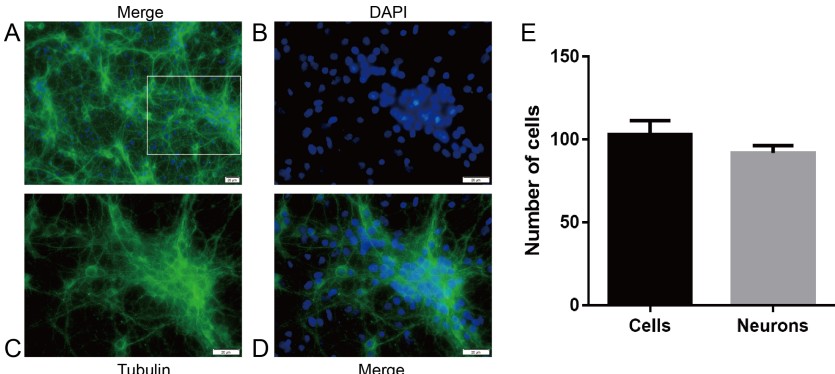

**Figure 2  The purity of hippocampal neurons from SD rats.** Primary cultured hippocampal neurons were cultured for 8 days before DNLA treatment. (A) hippocampal neurons 1,000×. (B) DAPI 2,000×. (C)Tubulin 2,000×. (D) hippocampal neurons 2,000×. (E) Neuron purity was more than 92.5% ($n = 3$). White squares represent enlarged areas.

anti-A$\beta_{1-42}$ (1:1,000), anti-$\beta$-actin (1:2,000), and anti-GAPDH (1:2,000) at 4 °C overnight, followed by incubation with secondary antibody at 4 °C for 1 h. The membranes were visualized using the chemiluminescence reagent ECL Plus (E003-100). The image was scanned, and band densities were quantified using Quantity One 1D analysis software v4.52 (Bio-Rad). GAPDH or $\beta$-actin was used to normalize protein loading.

## Statistical analysis
All data are expressed as the mean $\pm$ SD and were analyzed statistically by SPSS 17.0 software. The normally distributed data were first analyzed statistically via one-way analysis of variance (ANOVA). $P < 0.05$ was considered to be statistically significant.

## RESULTS

### The purity of hippocampal neurons in SD rats
First, we examined the purity of neurons by immunofluorescence staining to ensure the accuracy and feasibility of the experiment. The neuronal purity was at least 92.5% (Fig. 2).

### Effects of DNLA on the cell viability of hippocampal neurons in SD rats
Second, to determine whether DNLA improved the cell viability of hippocampal neurons from SD rats, we treated primary hippocampal neurons with DNLA for 48 h and detected the viability of neurons in each group by MTT assays. The results showed that DNLA did not significantly change the cell vitality of hippocampal neurons (Fig. 3). We concluded that DNLA does not alter the overall state of the cells in premature aging hippocampal neurons. However, did the internal state of the neurons change? We further examined this question based on the ''A$\beta$ cascade hypothesis''.

### DNLA decreased the accumulation of A$\beta$ by decreasing APP
As mentioned above, dementia is attributed to synaptic dysfunction and neuronal loss in the hippocampus and its associated cortex, which are caused by the accumulation of

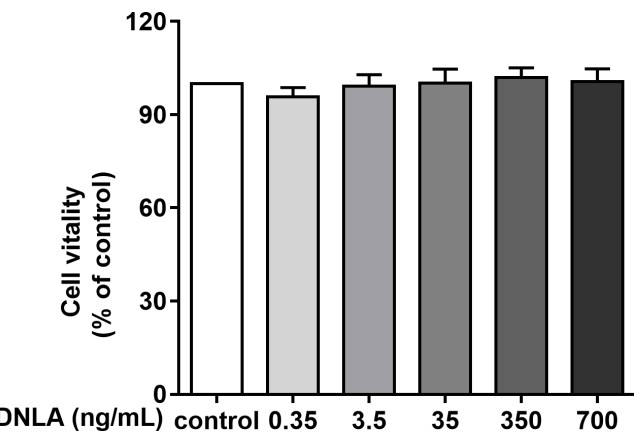

**Figure 3** **Effects of DNLA on the cell viability of hippocampal neurons from SD rats.** DNLA did not significantly change the cell vitality of hippocampal neurons ($n = 3$).

Aβ oligomers. Thus, we detected the protein expression of $Aβ_{1–42}$, and found that DNLA decreased the protein expression of $Aβ_{1–42}$ (Fig. 4B). To determine whether the effects of DNLA on the protein expression of $Aβ_{1–42}$ in hippocampal neurons of SD rats were related to APP, which is the source of Aβ and is cleaved to $Aβ_{1–42}$ and the other Aβ fragment, we examined the protein expression of APP in hippocampal primary neurons by Western blot analysis. Furthermore, PS1 (γ-secretase) participates in both the "non-amyloidogenic" and the "amyloidogenic" pathways. Therefore, we also examined the protein expression of PS1 by Western blot analysis. As shown in Fig. 4, DNLA decreased the protein expression of APP in hippocampal primary neurons of rats but did not change the protein expression of PS1. We found that DNLA decreased the accumulation of Aβ by decreasing APP.

## DNLA decreased the accumulation of Aβ by regulating α-secretase

As a protective factor for the progression of AD, α-secretase plays a significant role in the "non-amyloidogenic" pathway. Moreover, ADAM10 and ADAM17 are considered to be the most important α-secretases involved in the physiological processing of APP in the brain. We further probed the effect of DNLA on α-secretases (ADAM10 and ADAM17). And we examined the protein expression of ADAM10 and ADAM17 by Western blot analyses. Our results indicated that DNLA increased the protein expression of ADAM17 but decreased the protein expression of ADAM10 (Fig. 5). Thus, DNLA decreased the accumulation of Aβ by regulating α-secretase.

## DNLA decreased the accumulation of Aβ by inducing BACE1 (β-secretase)

In addition to γ-secretase, β-secretase, the crucial factor in the "amyloidogenic" pathways, is related to APP metabolism, which is the source of Aβ. BACE1 was shown to have important physiological roles in Aβ production. Inhibiting the activity of BACE1 can reduce the accumulation of Aβ, which triggers the exacerbation of AD. We confirmed that DNLA could decrease the production of Aβ. Moreover, the reduction of Aβ involved APP

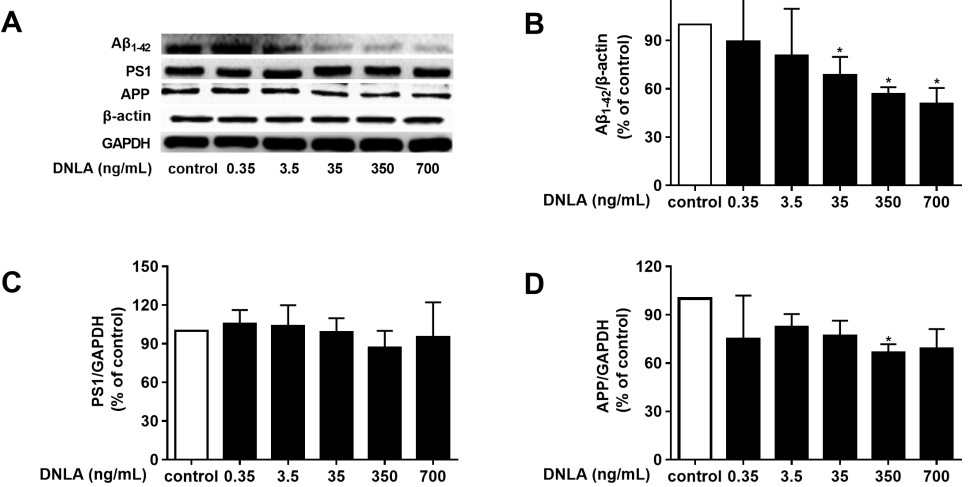

**Figure 4** **DNLA reduced the production of Aβ by decreasing APP.** The protein expression levels of Aβ$_{1-42}$ (B), PS1 (C) and APP (D) as determined from densitometric scans of Western blots. DNLA significantly decreased the protein expression of Aβ$_{1-42}$ (35, 350, 700 ng/mL) and APP (350 ng/mL). However, DNLA did not change the protein expression of PS1. (A) Representative strips of these proteins. Data are presented as the mean ± SD ($n = 4$). *$P < 0.05$ versus the sham group.

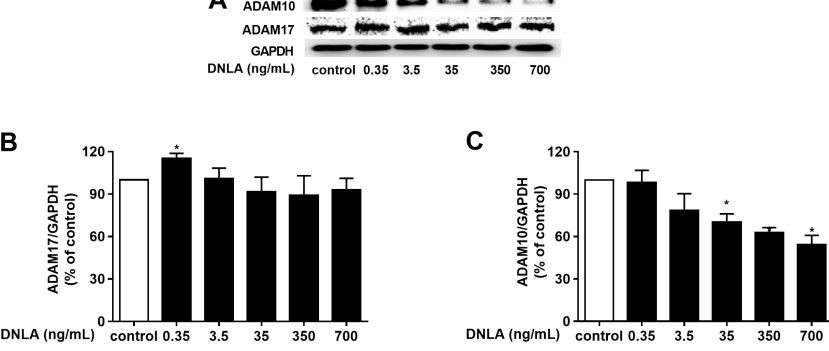

**Figure 5** **DNLA reduced the accumulation of Aβ through non-amyloidogenic pathways.** The protein expression levels of ADAM17 (B) and ADAM10 (C) as determined from densitometric scans of Western blots analyses. DNLA significantly decreased the protein expression of ADAM10 (35, 350, 700 ng/mL). However, this treatment significantly increased the protein expression of ADAM17 (0.35 ng/mL). (A) Representative strips of these proteins. Data are presented as the mean ± SD ($n = 4$). *$P < 0.05$ versus the sham group.

and α-secretase. Thus, we examined whether this effect was related to BACE1. Accordingly, we detected the protein expression of BACE1 by Western blots analyses. As indicated in Fig. 6, DNLA could decrease the accumulation of Aβ by reducing the protein expression of BACE1 in hippocampal neurons from SD rats.

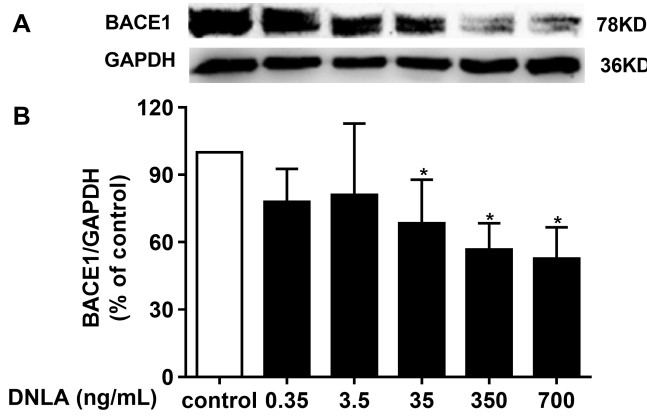

**Figure 6** **DNLA reduced the production of Aβ through amyloidogenic pathways.** Representative strip of the protein (A). The protein expression level of BACE1 (B) as determined from densitometric scans of Western blots. DNLA significantly decreased the protein expression of BACE1 (35, 350, 700 ng/mL). Data are presented as the mean ± SD ($n = 4$). $^*P < 0.05$ versus the sham group.

## DISCUSSION

DNLA was extracted from *D. nobile* Lindl, which is a traditional Chinese medicinal material and is produced in Chishui, Guizhou. As a pharmacologically active ingredient of *D. nobile* Lindl, DNLA has a protective effect on the nervous system. Previous studies showed that APP, β-secretase and Aβ protein in the hippocampus of Tg2576 transgenic mice were reduced. Additionally, DNLA substantially improved the learning and memory of middle-aged APP/PS1 transgenic mice and wild-type mice, suggesting that DNLA has potential health benefits (*Linshan et al., 2016*; *Nie et al., 2018*). In this experiment, neurons in the normal growth state were treated with different concentrations of DNLA, and the effects of DNLA on cell viability, APP, Aβ and its related secretases (α-, β- and γ-secretases) in normal conditions were preliminarily explored.

The senile plaque formed by extracellular deposition of Aβ is one of the main pathological characteristics of AD. Moreover, with increasing age, Aβ deposition increases (*Niedowicz et al., 2014*). Aβ mainly includes $Aβ_{1-40}$ and $Aβ_{1-42}$. $Aβ_{1-42}$ has stronger neurotoxicity and hydrophobicity than $Aβ_{1-40}$, and it has a strong tendency to oligomerize than $Aβ_{1-40}$ and is thus more easily polymerized in the brain. $Aβ_{1-42}$ accumulates to a high level, forming "senile plaques", which produce neurotoxic effects that injure the neurons and cause damage. Thus, inhibiting the production of $Aβ_{1-42}$ can delay the pathogenesis of AD (*Zhu et al., 2011*). In this study, the protein expression of $Aβ_{1-42}$ in hippocampal neurons of SD rats was detected. The results showed that DNLA can reduce the protein expression of $Aβ_{1-42}$ in hippocampal neurons.

APP is the precursor protein of Aβ and is a member of a family of related proteins, including amyloid precursor-like proteins (APLP1 and APLP2) in mammals and amyloid precursor protein-like (APPL) in *Drosophila*, all with large extracellular structures. These proteins have a one-way transmembrane domain, but only APP produces amyloidosis fragments (*O'Brien & Wong, 2011*). In theory, reductions in APP, as the precursor protein
of Aβ, could decrease the production of Aβ from the source. In this study, DNLA decreased the protein expression of APP.

Regardless, γ-secretase participates in the "non-amyloidogenic" and the "amyloidogenic" pathways and plays an indispensable role in the progression of APP metabolism to generate Aβ. Notably, DNLA reduced APP in a dose-dependent manner, but the effects were only significant at a dose of 350 ng/mL. One possible reason is that the data for these statistics are not sufficient to produce significant differences at other doses. Another possible reason is that the other dosages did not result in a significant change in this indicator. Simultaneously, some studies have suggested that the inhibition of γ-secretase activity could reduce the production of Aβ. Moreover, γ-secretase contains PS (including PS1 and PS2), nicastrin (NCT), PSEN enhancer 2 (Pen-2) and either anterior pharynx 1 (Aph-1) and is also the rate-limiting enzyme of APP production of Aβ (*Li et al., 2011a*). Many studies have shown that inhibition of γ-secretase activity helps reduce Aβ production (*Wolfe, 2012*). As the core catalytic subunit of γ-secretase, PS1 is the main active component (*Garcia-Ayllon et al., 2013*) of this enzyme and has an independent γ-secretase role (*Woodruff et al., 2013*). PS1 overexpression may be a risk factor for late-onset SAD (*Li et al., 2011a*). Conversely, inhibition of PS1 expression reduced Aβ production (*Futai et al., 2016*). In our study, the protein expression of PS1 showed no significant changes. These results indicated that DNLA did not affect PS1. $Aβ_{1-42}$ produced by γ-secretase did not increase and was reduced. Therefore, we concluded that DNLA affects APP metabolism to Aβ in other ways. We further explored this phenomenon.

Next, to explore the effect of DNLA on the non-amyloidogenic pathway, we detected the α-secretases, mainly the ADAM10 and ADAM17 proteins. ADAM17 is an important α-secretase involved in the non-amyloidogenic pathway of APP. Enhancing the activity of ADAM17 increased the secretion of a soluble sAPP α fragment with neuroprotective activity and decreased Aβ production. Therefore, ADAM17 is considered a potential therapeutic target for AD (*Qian, Shen & Wang, 2016*). The results of this experiment showed that DNLA could increase the protein expression of ADAM17 and reduce the production of Aβ. Additionally, ADAM10 also degrades APP by the non-amyloidogenic pathway. On one hand, it cleaves APP to produce sAPP α and avoids the production of Aβ. On the other hand, studies have also found an age-dependent increase in ADAM10 levels (*Schuck et al., 2016*). However, experiments by *Shackleton, Crawford & Bachmeier (2016)* suggested that inhibition of ADAM10 promoted the clearance of $Aβ_{1-40}$ and $Aβ_{1-42}$ in the brains of mice with AD, thereby reducing the level of Aβ. This result may be because the other α-secretase process APP when ADAM10 is reduced or absent. Other studies have reported that other members of the α-secretase family, such as ADAM9 and ADAM17, can compensate for the reduction in ADAM10 activity (*Asai et al., 2003*; *Hartmann et al., 2002*). When ADAM10 was absent or reduced, sAPP α production was shown to be reduced or Aβ was significantly increased, which may be due to the lack of compensation for α-secretase (*Kuhn et al., 2010*; *Postina et al., 2004*; *Suh et al., 2013*). In addition, inhibition of BACE1 increased ADAM10 cleavage of APP, but decreasing ADAM10 activity increased the risk of AD by increasing β-secretase cleavage of APP (*Colombo et al., 2013*). In this experiment, ADAM10 protein expression in hippocampal neurons treated with DNLA

was decreased. However, interestingly, ADAM17 protein expression was increased, and the Aβ protein level was decreased. These findings suggested that DNLA can reduce the expression of ADAM10 protein, which promotes the clearance of Aβ and reduces the Aβ level in hippocampal neurons of SD rats.

In the amyloidogenic pathway, APP finally produces Aβ by β- and γ-secretase cleavage. β-secretase, including BACE1, cathepsin S (Cat S), cathepsin L (Cat L) (*Schechter & Ziv, 2011*), cathepsin D (Cat D) and cathepsin B (Cat B) (*Zhou et al., 2012*), is the rate-limiting enzyme of APP production of Aβ. BACE2 was previously suggested to belong to the β-secretase family, but it was later proven to be a newly discovered η-secretase (*Willem et al., 2015*). Although Cat S, Cat L and Cat B may be further developed as targets for the treatment of AD (*Schechter & Ziv, 2011*), the current main target for β-secretase remains BACE1. Numerous studies have shown that inhibition of BACE1 can reduce the production of Aβ (*Adwan, Subaiea & Zawia, 2014*; *Yan & Vassar, 2014*; *Yun et al., 2013*; *Zhu et al., 2012*). The results of this experiment showed that BACE1 protein expression was decreased after treatment with DNLA, suggesting that DNLA can reduce the protein expression of BACE1 in neurons. This change likely caused the reduced protein expression of $Aβ_{1-42}$, which is crucial for the development of AD.

Under the experimental conditions, there are some drawbacks in our study. We need further direct evidence showing that DNLA reduces Aβ by affecting α- and β-secretase. For further analysis of the effects of DNLA on α-, β- and γ-secretase, the products of APP, such as sAPP α, P3, sAPPβ and CTF- γ, which were lysed by the corresponding secretases, should be further tested to explain the effect of DNLA on related secretases and its functions. In addition, the α-, β-, and γ-secretase activity should be directly detected for further confirmation of our results. Moreover, Aβ was reduced by decreasing its source and promoting its clearance. In this experiment, $Aβ_{1-42}$ protein expression in hippocampal neurons of SD rats was reduced, which was the result of various mechanisms. Therefore, the exact mechanism of $Aβ_{1-42}$ reduction requires further elucidation.

In summary, DNLA decreases Aβ accumulation by regulating α- and β-secretases in hippocampal neurons from SD rats. We verified that DNLA has potential health benefits and provided a theoretical basis for the anti-aging effect of DNLA.

## CONCLUSIONS

The results showed that DNLA can decrease $Aβ_{1-42}$ in hippocampal neurons of rats by regulating α- and β-secretase.

### Funding

This work was supported by the National Natural Science Foundation of China (No. 81473201), the Fund of Zunyi Medical University (No. F-900), the Fund of Zunyi Science and Technology Bureau & Zunyi Medical University (No. 2018-23), The Science and Technology Foundation of Guizhou Province of China, (No. JZ [2014]2016), and

Shijingshan's Tutor Studio of Pharmacology (No. GZS-2016-07). The funders had no role in study design, data collection and analysis, decision to publish, or preparation of the manuscript.

### Grant Disclosures

The following grant information was disclosed by the authors:

National Natural Science Foundation of China: 81473201.

Fund of Zunyi Medical University: F-900.

Fund of Zunyi Science and Technology Bureau & Zunyi Medical University: 2018-23.

The Science and Technology Foundation of Guizhou Province of China: JZ [2014]2016.

Shijingshan's Tutor Studio of Pharmacology: GZS-2016-07.

### Competing Interests

The authors declare there are no competing interests.

### Author Contributions

- Juan Huang and Nanqu Huang performed the experiments, analyzed the data, prepared figures and/or tables, authored or reviewed drafts of the paper, approved the final draft.
- Minghui Zhang performed the experiments, analyzed the data, prepared figures and/or tables.
- Jing Nie contributed reagents/materials/analysis tools.
- Yunyan Xu performed the experiments, contributed reagents/materials/analysis tools.
- Qin Wu conceived and designed the experiments, contributed reagents/materials/analysis tools.
- Jingshan Shi conceived and designed the experiments, authored or reviewed drafts of the paper, approved the final draft.

### Animal Ethics

The following information was supplied relating to ethical approvals (i.e., approving body and any reference numbers):

All animal procedures were approved by the Animal Experimental Ethical Committee of Zunyi Medical University [(2019)2-995].

### Data Availability

The raw measurements are available in the Supplemental Files.

### Supplemental Information

Supplemental information for this article can be found online at http://dx.doi.org/10.7717/peerj.7627#supplemental-information.

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
