# Peer review of "Dendrobium alkaloids decrease Aβ by regulating α- and β-secretases in hippocampal neurons of SD rats"

_PeerJ, doi:10.7717/peerj.7627_

## Round 0.1 · original submission · Major Revisions

Please revise the paper according to the comments.

[]

Reviewer 1 ·

Basic reporting

The current study by Huang et al. reports that DNLA decreases the Aβ through regulating α- and β-secretase of hippocampal neurons of rat. Although the study does raise some potentially interesting findings there are some limitations and the manuscript must be improved in several ways. Please find some concerns below:

In introduction: It appears to be quite superficial and brief on why the authors study DNLA in this experiment (just one paragraph). Suggest the authors expand the relevant background.

In Results: The authors missed information on the statistics, for example, the dose-dependent effects in Figs 4-6, and descriptions on some special results such as the the significant effect in 0.35 group but not in other groups in Fig 5B. There are no bars in the control columns in Figs 3-6? The authors should avoid describing the results as repetition of the figure legends.

In Discussion: Give some sentences on what is the significance of this study.

The authors should give GREATER attention to English grammars and punctuations, and the reference citations.

Experimental design

No comment.

Validity of the findings

No comment.

Additional comments

None.

Reviewer 2 ·

Basic reporting

The authors tried to demonstrate Dendrobium nobile Lindl. Alkaloids (DNLA) extracted from a Chinese medicinal herb decreased Aβ accumulation through regulating α- and β-secretase of hippocampal neurons in SD rat. Under current situation, there still lacks robust evidence to support their hypothesis.

Experimental design

The authors observed that DNLA reduced the protein expression of amyloid precursor protein (APP), disintegrin and metalloprotease 10 (ADAM10), β-site APP cleaving enzyme 1 (BACE1) and Aβ1-42 of hippocampal neurons in rat, increased the protein expression of ADAM17. However, they used an in vitro model of primary cultured NORMAL neurons! And declared that after cultured for 8-10 days in vitro, neurons were degenerated and accumulation of Amyloid beta peptide were observed. I don not think this is the common phenotype of NORMAL primary cultured neurons under such a circumstances and this treatment is also not a valid model for studying the clearance or degradation of Aβ. Please show more evidence to support their experimental design here and make some modifications.

Validity of the findings

As I questioned above, the authors used an uncommon model to test their hypothesis. Thus, the validity of their findings is doubtful.

Additional comments

The manuscript suffers from many typos and poor grammar throughout. As they are currently written, many sentences are confusing. The manuscript would greatly benefit from a thorough editing by a native English speaker.

---

## Round 0.2 · accepted · Accept

Thanks for the efforts in improving the manuscript,

Reviewer 1 ·

Basic reporting

The authors have addressed the issues, and improved the quality of the ms.

Experimental design

No comment.

Validity of the findings

No comment.

Additional comments

It is OK now.